# Genomic Divergence Characterization and Quantitative Proteomics Exploration of Type 4 Porcine Astrovirus

**DOI:** 10.3390/v14071383

**Published:** 2022-06-24

**Authors:** Jie Tao, Benqiang Li, Jinghua Cheng, Ying Shi, Changtao Qiao, Zhi Lin, Huili Liu

**Affiliations:** 1Institute of Animal Husbandry and Veterinary Medicine, Shanghai Academy of Agricultural Sciences, Shanghai 201106, China; livia_taojie@126.com (J.T.); libenqiang2007@163.com (B.L.); zero5cheng@163.com (J.C.); shiyingsunny@126.com (Y.S.); qiaochangtao@163.com (C.Q.); linzhiplain@163.com (Z.L.); 2Shanghai Key Laboratory of Agricultural Genetic Breeding, Shanghai 201106, China; 3Shanghai Engineering Research Center of Pig Breeding, Shanghai 201302, China

**Keywords:** porcine astrovirus, ORF2, genomic characterization, divergence time, proteomics

## Abstract

Porcine astrovirus (PAstV) has been identified as an important diarrheic pathogen with a broad global distribution. The PAstV is a potential pathogen to human beings and plays a role in public health. Until now, the divergence characteristics and pathogenesis of the PAstV are still not well known. In this study, the PAstV-4 strain PAstV/CH/2022/CM1 was isolated from the diarrheal feces of a piglet in Shanghai, which was identified to be a recombination of PAstV4/JPN (LC201612) and PAstV4/CHN (JX060808). A time tree based on the ORF2 protein of the astrovirus demonstrated that type 2–5 PAstV (PAstV-2 to 5) diverged from type 1 PAstV (PAstV-1) at a point from 1992 to 2000. To better understand the molecular basis of the virus, we sought to explore the host cell response to the PAstV/CH/2022/CM1 infection using proteomics. The results demonstrate that viral infection elicits global protein changes, and that the mitochondria seems to be a primary and an important target in viral infection. Importantly, there was crosstalk between autophagy and apoptosis, in which ATG7 might be the key mediator. In addition, the NOD-like receptor X1 (NLRX1) in the mitochondria was activated and participated in several important antiviral signaling pathways after the PAstV/CH/2022/CM1 infection, which was closely related to mitophagy. The NLRX1 may be a crucial protein for antagonizing a viral infection through autophagy, but this has yet to be validated. In conclusion, the data in this study provides more information for understanding the virus genomic characterization and the potential antiviral targets in a PAstV infection.

## 1. Introduction

The astrovirus (AstV) is a small, non-enveloped, positive-sense single-stranded RNA virus which belongs to the family *Astroviridae*, with a genome of 6–8 kb in length [1]. It can infect a wide range of hosts from birds to mammals, including humans, causing clinical symptoms such as diarrhea, vomiting, and virus-associated hepatitis in birds, or encephalitis in humans and other mammals [2]. This virus is the second most common cause of human diarrhea, with great variability and host adaptations [3]. Some studies indicate that genome recombination exists among different serotypes or species, which means that there may be inter-species transmission between animals and between animals and humans [4].

Porcine astrovirus (PAstV), belonging to the genus *Mamastrovirus*, has been identified as a potentially important diarrheic pathogen [5]. Since its first isolation in 1980, the PAstV has been found to be prevalent in many countries including the USA, Canada, Italy, Korea, and Japan [6,7,8,9,10]. In China, the PAstV is widely prevalent, with a positivity rate of approximately 7.5–46.3% [11]. To date, five PAstV genotypes have been identified in various countries [12]. PAstVs isolated in different provinces of China possess varied genotypes and prevalence. All five PAstV genotypes are present in Guangxi [13]; PAstV-1, -2, -4, and -5 in Hunan [14,15]; PAstV-2 and -5 in Sichuan [16] and Yunan [11]; PAstV-5 in Jilin [17]; PAstV-4 in Anhui [18] and Tianjin [19]; and PAstV-2 in Shanghai [20]. However, to date, information available on the pathogenesis of PAstV is limited.

Our research team has focused on the epidemiological surveillance of porcine diarrhea pathogens for more than ten years. We found that the infection patterns of diarrheal pathogens have changed greatly. Porcine epidemic diarrhea virus (PEDV), transmissible gastroenteritis virus (TGEV), and porcine rotavirus (PoRV) were the three dominant pathogens in diarrheal disease ten years ago. Nevertheless, the prevalence of several underappreciated diarrheic pathogens, such as PAstV, porcine sapello virus (PSV), and porcine kobuvirus (PKoV), are increasing now while that of TGEV and PoRV have dwindled [21]. Importantly, these underappreciated diarrheic pathogens often co-infect with PEDV in clinical samples and play synergistic pathogenic roles [22]. This indicates that we need to pay attention to the pathogenic mechanism of these pathogens. Moreover, PAstV has the potential for interspecies transmission, which is of great public health significance. Therefore, we consider PAstV to be one of the most significant diarrheic pathogens apart from PEDV.

The identification and genetic analysis of new variants are useful for understanding viral evolution, and new vaccine strategies are necessary for the efficient control of enteric viruses. Up to now, PAstV had been reported in nearly all provinces of China. However, there are only two available complete genome sequences of PAstV strains isolated in Shanghai: PAstV-1 strain Shanghai 2008 (GQ914773) [23] and PAstV-2 strain JWH-1 (HQ637283) [20]. In this study, we identified a PAstV-4 strain in Shanghai, China. Its genetic characteristics and divergence time were parsed, and the infection mechanism explored using LC-MS/MS quantitative proteomics technology. Several crucial proteins, biological processes, and KEGG pathways were explored, which will be of great use in follow-up mechanism studies.

## 2. Materials and Methods

### 2.1. Virus Isolation and Identification

Five fecal samples were collected from piglets with diarrhea and then detected by RT-PCR (Appendix A) [24]. A porcine astrovirus-positive fecal sample was treated with penicillin–streptomycin solution (HyClone, Thermo, Waltham, MA, USA) overnight at 4 °C. The liquid was filtered using 0.22 μm Millex (Millipore^®^, Burlington, MA, USA), and inoculated in porcine kidney 15 (PK15) cells, with the addition of 15 μg/mL pancreatin (Sigma Aldrich, St. Louis, MO, USA). The cell line PK15 was obtained from the China Center for Type Culture Collection (Wuhan, Hubei, China). A total of 72 h later, cell culture was collected following three rounds of freeze–thaw. Finally, the fifth-passaged virus (PAstV/SH/2022/CM1) was identified by both RT-PCR and an indirect immunofluorescence assay.

### 2.2. Immunofluorescent Staining

PK15 cells were fixed with 4% paraformaldehyde (PFA) (Takara^®^, Beijing, China) at room temperature for 30 min. Then, cells were washed with PBS, followed by permeabilization, and blocking with PBST buffer (0.2% Triton X-100 in PBS buffer) containing 5% normal goat serum for 30 min at room temperature. PAstV monoclonal antibody (LVDU, Shandong, China) was diluted with PBST buffer (1:500). Cells were immunostained with primary antibodies at 37 °C for 1 h, and then immunostained with FITC-labeled sheep anti-mouse IgG secondary antibody (1:2000) (Invitrogen, Carlsbad, CA, USA) at 37 °C for 1 h. Finally, cells were counterstained with ProLong^TM^ Diamond Antifade Mountant (Invitrogen, Carlsbad, CA, USA) and imaged using a Zeiss microscope.

### 2.3. Full-Length Genome Sequencing of PAstV/SH/2022/CM1

The full-length genome of PAstV isolate was divided into six fragments (Appendix A), which were amplified by RT-PCR and cloned into the vector pEASY Blunt Zero (TransGen, Beijing, China) in accordance with the manufacturer’s protocol. The 5′- and 3′-end sequences were obtained using SMARTer RACE 5′/3′ Kit) (Takara^®^, Beijing, China). The PCR products were sent to BioSune Biotechnology Co., Ltd. (Shanghai, China) for sequencing. All of the sequencing reactions were done in quintuplicate, and all sequences were confirmed by sequencing both strands.

Sequence analysis was conducted using the EditSeq tool included with the Lasergene DNASTAR^TM^ 7.0 software package (DNASTAR Inc., Madison, WI, USA).

### 2.4. Genomic Characterization

To evaluate the genetic relationships and evolution rate of the PAstVs, ORF2 sequences of the available PAstVs retrieved from GenBank were analyzed by a phylogenetic tree and time tree using the maximum likelihood method, implemented in MEGA X [25] with 1000 bootstrap replicates (Appendix A). The cut-off value for the condensed tree was 50%. The divergence time was estimated for all branching points in a tree using the RelTime with Dated Tips (RTDT) method [26]. A total of 41 PAstV strains identified from 1983 to 2022 were referred to in this analysis. Furthermore, recombination analysis was performed using Recombination Detection Program (RDP) software version 4 based on the whole genome [27].

### 2.5. Protein Preparation and Liquid Chromatography–Tandem Mass Spectrometry (LC–MS/MS) Analysis

PK15 cells were infected with 5 MOI of PAstV/SH/2022/CM1 and then collected at 24 hpi, as well as uninfected PK15 cells, with a cell scraper, followed by centrifuging at 1500× *g* for 10 min and then washing with PBS. There were three replicates per sample. The collected cells were sonicated three times on ice in lysis buffer (8 M urea, 1% protease inhibitor cocktail). Cell debris was removed by centrifugation at 12,000× *g* at 4 °C for 10 min. For tryptic digestion, the protein solution was reduced with 5 mM dithiothreitol for 30 min at 56 °C and alkylated with 11 mM iodoacetamide for 15 min at room temperature in darkness. Then, the protein was diluted by adding 100 mM TEAB to urea (less than 2 M). Finally, trypsin was added at a 1:50 trypsin-to-protein mass ratio for the first digestion overnight and a 1:100 trypsin-to-protein mass ratio for a second 4 h digestion.

### 2.6. LC–MS/MS Analysis with 4D Mass Spectrometer

The tryptic peptides were dissolved in solvent A (0.1% formic acid, 2% acetonitrile/in water), directly loaded onto a homemade reversed-phase analytical column. Peptides were separated with a gradient of solvent B (0.1% formic acid in acetonitrile) from 6% to 24% over 70 min, 24% to 35% over 14 min and climbing to 80% over 33 min, then holding at 80% for the last 3 min, all at a constant flow rate of 450 nL/min on a nanoElute UHPLC system (Bruker Daltonics).

The peptides were subjected to a capillary source followed by the timsTOF Pro (Bruker Daltonics) mass spectrometry. The electrospray voltage applied was 1.60 kv. Precursors and fragments were analyzed at the TOF detector, with a MS/MS scan range from 100 to 1700 *m*/*z*. The timsTOF Pro was operated in parallel accumulation serial fragmentation (PASEF) mode. Precursors with charge states 0 to 5 were selected for fragmentation, and 10 PASEF-MS/MS scans were acquired per cycle. The dynamic exclusion was set to 30 s.

### 2.7. Database Search

The resulting MS/MS data were processed using the MaxQuant search engine (v.11.6.15.0). Tandem mass spectra were searched against the human SwissProt database (20,422 entries) concatenated with the reverse decoy database. Trypsin/P was specified as a cleavage enzyme, allowing up to 2 missing cleavages. The mass tolerance for precursor ions was set as 0.02 Da. Carbamidomethyl on Cys was specified as the fixed modification, and acetylation on the protein N-terminal and oxidation on Met were specified as variable modifications. FDR was adjusted to <1%.

### 2.8. Bioinformatics Analysis

All differential proteins were mapped to each term for annotation. Gene Ontology (GO) annotation was performed based on the UniProt-GOA database (http://www.ebi.ac.uk/GOA/, accessed on 26 January 2022). Protein domains were annotated by InterProScan based on the InterPro database (v75.0) (http://www.ebi.ac.uk/interpro/, accessed on 26 January 2022) with default parameters. The Kyoto Encyclopedia of Genes and Genomes (KEGG) pathway database (https://www.genome.jp/kegg/pathway.html, accessed on 26 January 2022) was used to annotate the protein pathway with the KEGG online service tool, KAAS (https://www.genome.jp/kaas-bin/kaas_main, accessed on 26 January 2022) and KEGG mapper (http://www.genome.jp/kegg/mapper.html, accessed on 26 January 2022). Sub-cellular localization predication was performed based on the UniProtKB database (https://www.uniprot.org/uniprot, accessed on 26 January 2022).

The differentially expressed proteins (DEPs) between the experimental group and the control group (*p* < 0.05, FC (fold change) >1.2 or <0.67) were identified according to the quantitative values obtained by LFQ intensity. All DEPs belonging to different sub-cellular compartments were searched against the STRING database version 11.0 (https://string-db.org/, accessed on 26 January 2022) to retrieve their interactions. Here, we selected all interactions with a confidence score >0.7 (high confidence). The interaction network from the STRING database was visualized using the Cytoscape 3.8.0 software.

## 3. Results

### 3.1. Identification and Genomic Characteristics of a PAstV4 Isolate

A PAstV positive sample was screened from the five clinical fecal samples by RT-PCR detection and then virus isolation was carried out. The PAstV isolate named as PAstV/SH/2022/CM1 was identified in PK15 cells by continuous passage culture, with the addition of 15 μg/mL pancreatin. The obvious shrivel and shed of PK15 cells appeared 60 h post-infection since the third generation (Figure 1a). The indirect immunofluorescence assay confirmed that the monoclonal antibody against PAstV could recognize the virion with a specific green fluorescence in the cytoplasm (Figure 1b). Furthermore, the whole genome of the 5th-passaged PAstV/SH/2022/CM1 was sequenced. It was 6660 nucleotides in length, including 5′UTR, ORF1a, ORF1b, ORF2, and 3′UTR. There was a 70 nt overlap between ORF1a and ORF1b containing a ribosomal frameshifting signal (AAAAAAC). A highly conserved sequence (U_4066_UUGGAGGGGCGGACCAAAN_11_) was linked before the initiation codon of ORF2, which could be part of the promoter for subgenomic RNA synthesis (Figure 1c).

### 3.2. Phylogenetic and Recombination Analysis

A topologically consistent phylogenetic tree was obtained, in which the ORF2 proteins of the most available Chinese PAstV strains were analyzed (Figure 2). The ORF2 gene sequence of PAstV/SH/2022/CM1 was submitted to GenBank (ON391462). The phylogenetic tree showed that PAstV-4 was the most common genotype and that there are currently no domestic PAstV-3 strain (Figure 2a). The ORF2 of PAstV/SH/2022/CM1 was most closely related to the Chinese strain JXJA (KX060808) isolated in Jiangxi in 2014.

To identify the possible recombination events in PAstV/SH/2022/CM1 genome, the RDP software was used. The results demonstrate that PAstV/SH/2022/CM1 is a recombinant of the PAstV4/JPN/MoI2-3-2/2015 (LC201612) and JXJA (KX060808) strains (Figure 2b). The RDP software identified two recombination breakpoints at 4970 bp and 6393 bp, indicating the recombinant sequence located in ORF2 gene. The region (nt 4970–6393) was closely related to the PAstV4/JPN/MoI2-3-2/2015 strain.

A temporal tree can be helpful to better understand the viral genetic evolution. A time tree using the RTDT method based on the ORF2 protein was calculated to investigate the divergence time of PAstVs. The tree revealed that PAstV-1 had the earliest origin among the porcine astroviruses, traced back to the 1960s. The other four PAstV genotypes were differentiated between 1997 and 2022, and the divergence time of PAstV-5 was the latest, around 2005 (Figure 2c). Furthermore, HAstV might have diverged from PAstV-1 around 1970, suggesting that interspecies transmission between humans and pigs may have occurred in the 1970s.

### 3.3. Defining the Proteomic Response of Host Cells to PAstV4 Infection

Considering the changes in host response to a viral infection at early phase, we collected the whole cell lysates of PK15 cells at 24 hpi. In this study, the differentially expressed proteins (DEPs) were detected and their relative rates of changes were analyzed. Totally, 6456 proteins were identified, of which 5125 proteins were quantifiable. The results showed PAstV/SH/2022/CM1 infection-induced broad modulations of the proteomes in PK15 cells. For the identification of DEPs, the cutoffs for the fold change in abundance and *p*-value were set to 1.2/0.67 and 0.05, respectively. Among the DEPs, 375 proteins (207 upregulated proteins and 168 downregulated proteins) were significantly modulated in the infected PK15 cells (Figure 3a).

### 3.4. Functional Characterization of the DEPs Based on GO and KEGG Annotation

Firstly, DEPs were analyzed based on GO categories, including cellular components (CCs), molecular function (MF), and biological process (BP). The results showed that GO terms were enriched following viral infection (Figure 3b). The COG/KOG category of cellular processes and signaling indicated that signal transduction mechanisms, post-translational modification, protein turnover, and chaperones were mostly enriched (Figure 3c). Among the top 20 enriched GO terms, the membrane and vesicle of CCs were highly enriched (Figure 4a). MFs mainly focused on binding and catalytic activity (Figure 4b). BPs were associated with the cellular process, with biological regulation and metabolism being particularly enriched (Figure 4c). To achieve different specific biological functions, gene products need to coordinate with each other in an orderly manner. Further, the KEGG pathway annotation was enriched in transcriptional misregulation, phosphatidylinositol signaling system, cellular senescence, and others (Figure 4d).

### 3.5. Mitochondria Might Be Most Heavily Affected Organelles for PAstV/SH/2022/CM1 Infection in Host Cells

The statistical analysis of the subcellular localization proportion of the different proteins of each comparison pair was carried out. The results show that 24 h after PAstV/SH/2022/CM1 infection, 32.27% of the DEPs were localized in the nucleus, 25.07% were localized in the cytoplasm, 13.87% were localized in the mitochondria, 10.67% were localized extracellularly, and 10.4% were localized in the plasma membrane (Figure 5a). It was indicated that mitochondrion might be the most heavily affected organelle for PAstV/SH/2022/CM1 infection in PK15 cells. A further Venn diagram based on the several important biological progresses indicated that Uracil-DNA glycosylase (UNG) and NOD-like receptor X1 (NLRX1) in mitochondrion play roles in the immune and antiviral response (Figure 5b). Vacuolar protein-sorting-associated protein 25 (VPS25), myeloid cell leukemia-1 (MCL1), and phosphatidylinositol 5-phosphate 4-kinase type-2 beta (PIP4K2B) participated in the biological progress of autophagy (Figure 5b).

Furthermore, we found that there were eight DEPs that participated in the defense response to virus infection, including six upregulated proteins and two downregulated proteins (Figure 5c) (Appendix A). Of these proteins, NLRX1 was located in the mitochondria, ATG7 located in the peroxisome, interferon regulator factor 7 (IRF7) and chemokine (C-X-C motif) ligand 10 (CXCL10) located in the extracellular space, and the other four proteins were located in the nucleus (Figure 5d).

In view of the close relationship of NLRX1, VPS25, and PIP4K2B with mitophagy, we speculate that PAstV/SH/2022/CM1 infection may cause mitochondrial injury—mitophagy—to resist the innate host immunity.

### 3.6. PPIs in the Immune and Antiviral Response

In the present work, heat-maps were generated on all filtered proteins. The results showed 39 DEPs within the immune and antiviral response, including 23 upregulated proteins and 16 downregulated proteins (Figure 6a) (Appendix A). There were four DEPs (ATG9A, ATG7, EIF2AK4, and PIK3R2) involved in autophagy while three DEPs (CTSZ, CTSH, and PIK3R2) were involved in apoptosis (Figure 6b), both of which were two common biological processes in the antiviral response. Additionally, proteins with the ubiquitin conjugation effect (NOS2, RPL10, RPL35, and UBA52) were also discovered. PPI revealed that many proteins had complicated interactions with other proteins, mainly involved in the Toll-like signaling pathway, NOD-like signaling pathway, RIG-I-like receptor signaling pathway, TNF signaling pathway, and IL-17 signaling pathway (Figure 6b). Interestingly, NLRX1, CXCL8, CXCL10, and IRF7 co-participated in the RIG-I-like receptor signaling pathway. Furthermore, we also found that CXCL family members, including CXCL3, CXCL9, CXCL10, and CXCL11, might interact with viral proteins (Figure 6c), which is a vital clue for virus–host interaction research.

Finally, the PPI of the DEPs involved in the autophagy, apoptosis, and defense response to the virus was a verification of the Venn analyses in Figure 5b. It was clearly divided into three clusters, with the ATG7 protein located in the center (Figure 6d). Here, we tentatively speculate that ATG7 played an important role in antagonistic PAstV infection according to the crosstalk between autophagy and apoptosis, which needs to be further verified.

## 4. Discussion

The PAstV has played an active role in pigs in the evolution and ecology of the Astroviridae [20]. Exploring its genetic evolution characteristics is the first and most necessary step. In the phylogenetic tree based on ORF2 sequences, the most available Chinese PAstVs were retrieved from GenBank and analyzed. Evidently, the PAstV-4 was the main genotype prevalent in China, and no PAstV-3 strain had been previously identified in China. It has been reported that the PAstV-3 is associated with polio encephalomyelitis/encephalitis [28]. In China, the PAstV is known primarily for causing diarrhea, which may be the foremost reason for the rarely identified PAstV-3 in domestic environments. Hence, surveillance should be strengthened in terms of the astrovirus infection with extraintestinal manifestations in the future.

Recombination events and mutations are the major factors that determine the molecular evolution of RNA viruses. Thus, it is important to consider what is known about astrovirus recombination in the context of different species [29]. There is increasing evidence of multiple recombination events between distinct PAstV strains and between PAstV and HAstV [30], as well as interspecies recombination, suggesting that cross-species transmission is frequent in astroviruses [8]. Previous studies suggested that PAstV-1, PAstV-2, PAstV-3, and PAstV-5 may have been transmitted across host species [14,31]. In this study, the strain PAstV/SH/2022/CM1 had experienced a recombination event with PAstV4/JPN (LC201612) and PAstV4/CHN (JX060808). Zhao et al. reported that PAstV4/Tianjin/2018 was a novel recombinant of PAstV4/US-IL135 (JX556692) and PAstV4/JPN (LC201608) [19]. It was confirmed that the recombinations of PAstV-4 occurred among the same species. The PAstV-4 strains prevalent in China probably originated from the strains circulating in neighboring countries. Pathogen surveillance of epidemic strains in neighboring countries should be strengthened, especially in Japan.

There is a growing interest in using morphological data to infer species divergence times in systematics by using morphological clock analyses [32], which is an excellent biological tool for delving deeper into genetic evolution. Information on the genetic evolution of the PAstVs is quite rare. Therefore, we first inferred the divergence time of astroviruses. The results showed that the divergence time of PAstV-1 was earliest among the five PAstV genotypes. Interestingly, HAstV might have diverged from PAstV-1 around 1970s, while the divergence time of the variant HAstV-MLB was between that of PAstV-2 and PAstV-5. Again, this proves that there is a risk of interspecies transmission of PAstV.

Research on the viral infection characteristics and pathogenesis of the PAstV are limited as yet. The virus itself is not an independent living entity, and it relies heavily on host machinery to achieve replication and spread [33]. Proteomics is an effective technology to explore rich research fields in preliminary studies. To gain a comprehensive, unbiased overview of host responses in terms of global proteins to PAstV infection, quantitative proteomic analysis by LC–MS/MS and advanced bioinformatics was performed in this study. Here, we adopted a quantitative, label-free proteomics approach based on tapped ion mobility spectrometry (TIMS) time-of-flight (TOF) pro, introduced by Bruker Inc. in 2017. This approach utilizes a quadruple parallel accumulation serial fragmentation acquisition method by synchronizing an MS/MS precursor selection with a TIMS separation, which can increase the overall depth and resolution of the proteomic data with small samples.

Proteomic analysis in this study revealed that viral infection elicited a broad spectrum of protein changes in PK15 cells across many organelles, especially mitochondria. Given the vital role in energy metabolism, as well as protein and lipid synthesis, the disturbance of mitochondria may significantly affect the host cells. Huang et al. reported that swollen mitochondria, broken mitochondrial ridges, autophagosomes, and autophagolysosomes could be observed in goose nephritic astrovirus (GNAstV) infection [34]. This is consistent with the proteomic results in our study. Eukaryotes use autophagy to turn over organelles, protein aggregates, and cytoplasmic constituents. The impairment of autophagy causes developmental defects, starvation sensitivity, the accumulation of protein aggregates, neuronal degradation, and cell death [35]. In this study, we found MCL1 and PIP4K2B proteins upregulated after PAstV/SH/2022/CM1 infection, which were related to autophagy. MCL1 was a mitophagy receptor [36] and PIP4K2B could positively regulate autophagosome assembly [37]. In addition, the NLRX1 located in the mitochondria positively regulated the antiviral response through RIG-I like receptor signaling pathway and NOD-like signaling pathway, which also were related to autophagy [38,39]. The NLRX1, a mitochondria-targeted protein, is known to negatively regulate innate immunity and cell death responses [40]. It is also a candidate modulator of the interplay between mucosal inflammation, metabolism, and the gut microbiome during inflammatory bowel disease (IBD) [41]. The NLRX1 is highly expressed in the intestine, and known to modulate ROS production, mitochondrial damage, autophagy and apoptosis [42]. Li et al. found that NLRX1 could regulate mitophagy via the FUNDC1-NIPSNAP1/NIPSNAP2 signaling pathway [42]. Thus, it is foreseeable that NLRX1 may be a crucial protein in the host’s antiviral response against porcine astrovirus infection.

Ubiquitination is a widespread post-translational modification that controls multiple steps in autophagy [43]. In our study, we found two evolutionarily-conserved ubiquitin-like conjugation systems. The first one involved ATG7 and ATG10; the second one involved ATG3 and ATG7, which was consistent with the literature report [43]. Ubiquitin proteins are encoded by four genes (UBA52, UBA80, UBB, and UBC). UBA52 comprise a single ubiquitin fused at the C-terminus to ribosomal protein (RP) L40. UBA52 ubiquitination plays an important role between mitophagy and apoptosis [44]. PPIs in this study showed that ATG7-UBA52 ubiquitination participated in apoptosis while ATG7 also took part in autophagy. The functional relationship between apoptosis and autophagy is complex. It may be triggered by common upstream signals, and sometimes this may result in combined autophagy and apoptosis; in other instances, the cell switches between the two responses in a mutually exclusive manner [45]. As a consequence, we suggest that PAstV infection caused a crosstalk between autophagy and apoptosis, in which ubiquitination became involved.

Another worthy topic in this study is that several CXCL family members may have the potential to interact with viral proteins. It was reported that hepatitis C virus (HCV) core protein could induce CXCL10 expression through NF-κB signaling pathway in macrophages [46]. CXCL10 is a marker of host immune response, which is dependent on interferon (IFN)—γ [47]. CXCL10/IFN– γ-induced protein 10 (IP-10) has a direct action in control of dengue viral (DENV) [48] and Zika viral (ZIKV) replication [49]. In general, an early immune response to viral infection may determine its clinical manifestation and outcome. So, attenuation to the role of CXCL in PAstV infection is expected to reveal new antiviral targets.

To summarize the results, a PAstV-4 strain was identified in Shanghai, China, and the divergence time characteristics of astroviruses were highlighted. Furthermore, the proteomics analysis of astroviruses was carried out for the first time and a preliminarily revealed the important proteins, biological processes, and KEGG pathways in a PAstV-4 infection, providing effective clues for subsequent pathogenetic research. However, there is an imperfection point in this study. It’s well known that cell line selection is a vital factor in proteomics research. Porcine astrovirus is an enterovirus and the intestinal cells should be used to study the interaction mechanism between the PAstV and the host. However, PK15 cell is the only cell line used for the virus isolation of PAstV. In this study, some DEPss such as VPS25—which are often expressed in the kidney—may not really reflect the pathogenic mechanisms of the PAstV. We think this is an inevitable flaw in the initial study of new pathogens, especially in the absence of a suitable cell line. We will verify the proteomic data in vivo in follow-up studies. We will also try to engineer intestinal cell lines to accommodate porcine astrovirus proliferation.

## Figures and Tables

**Figure 1 viruses-14-01383-f001:**
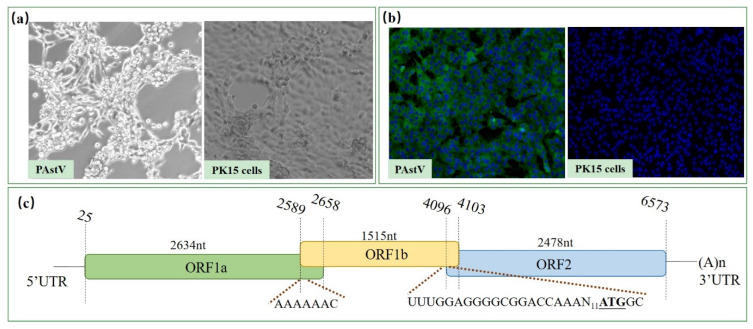
Identification and genomic structure of PAstV/SH/2022/CM1. (**a**) Cytopathic effect of the PAstV/SH/2022/CM1 isolate. The virus was inoculated in PK15 cells and cells appeared shriveled and shed at 60 hour (h) after infection. (**b**) Indirect immunofluorescence assay was carried out 48 h post-infection using the monoclonal antibody against PAstV, which indicated that specific green fluorescence could be detected in the cytoplasm. (**c**) Genomic organization, with three open reading frames: ORF1a, ORF1b, and ORF2.

**Figure 2 viruses-14-01383-f002:**
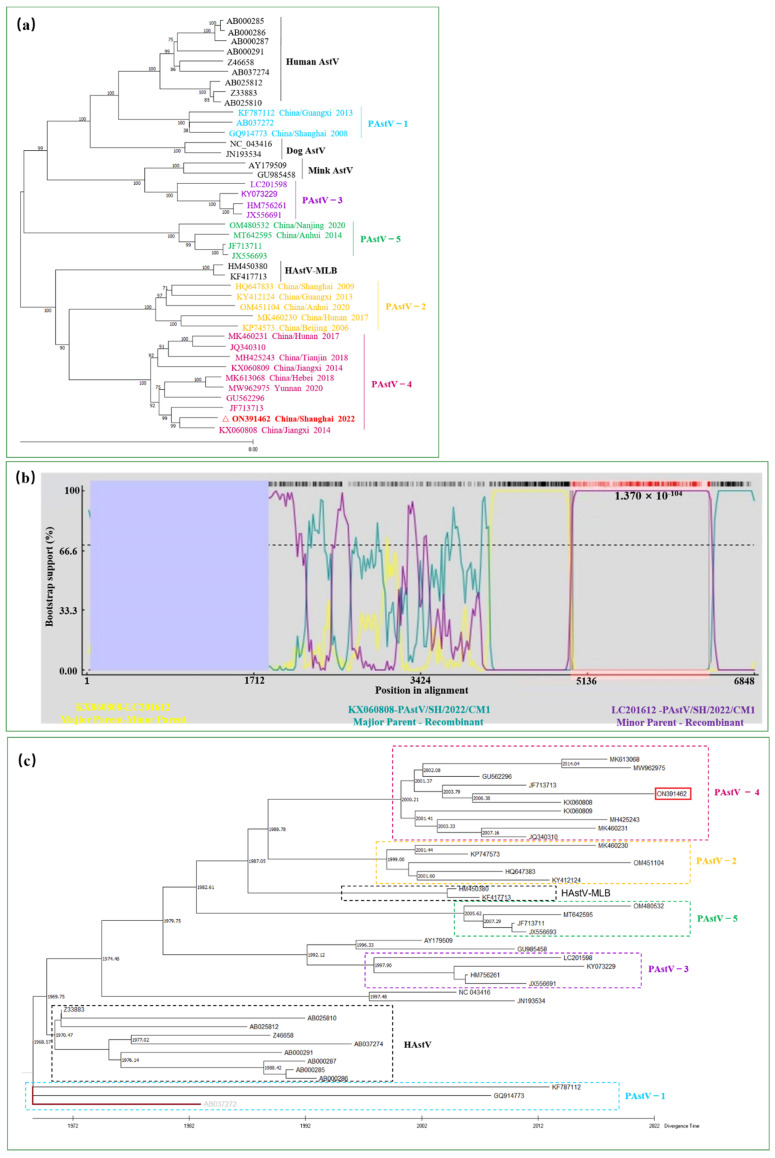
Genetic evolutionary analysis of PAstV/SH/2022/CM1. (**a**) A phylogenetic tree was constructed based on the ORF2 sequences of astroviruses using the maximum likelihood method implemented in MEGA X. (**b**) Viral recombination was further analyzed and the potential recombinant breakpoints were identified by RDP4.0. BOOTSCAN on the basis of pairwise distance with 100 bootstrap replicates. (**c**) The time tree was calculated in MEGA X where divergence time was inferred by the RelTime with Dated Tips (RTDT) method. The PAstV-1 strain (AB032272) (grey color) was designated as an outgroup taxon and all sequences used the year of sampling dates as the tip dates for calibration constraints. The divergence time of each branch is marked. PAstV-1, PAstV-2, PAstV-3, PAstV-4, and PAstV-5 strains are circled in different colors of dotted boxes.

**Figure 3 viruses-14-01383-f003:**
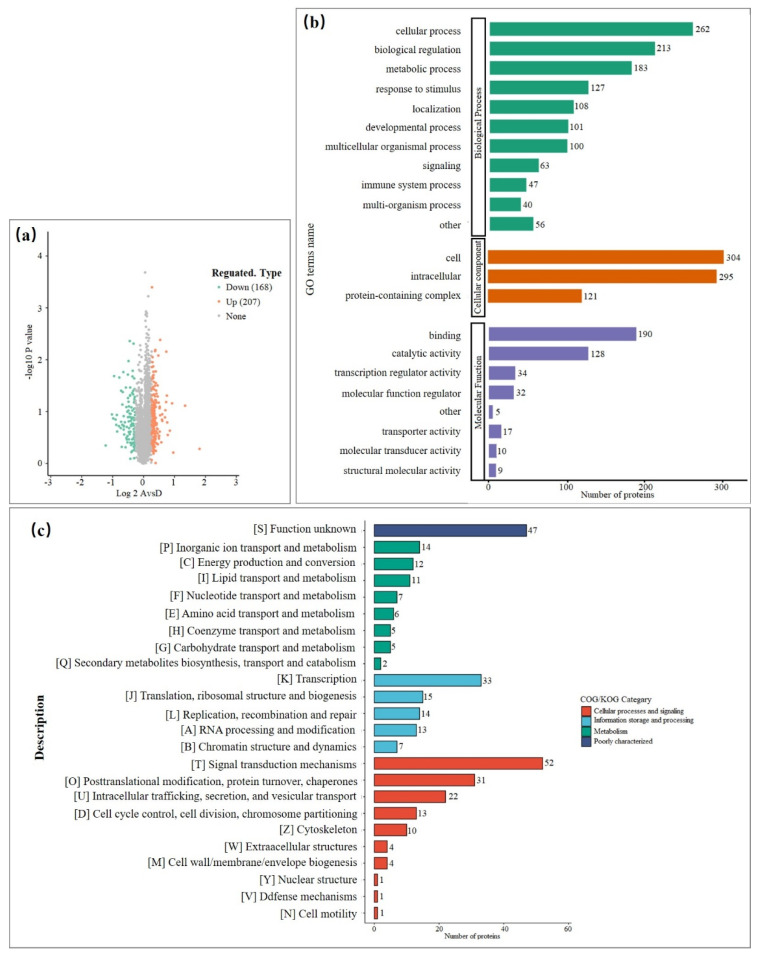
Proteomic analysis of PK15 cells following PAstV/SH/2022/CM1 infection. (**a**) Volcano plots show the regulated proteins of cells following PAstV infection. Proteins differentially expressed (DEPs) with fold change over 1.2 and *p* < 0.05 are marked in color. *p*-values were calculated using a two-sided, unpaired Student’s *t* test with equal variance assumed (*n* = 3 independent biological samples). (**b**) Gene Ontology (GO) enrichment analysis of DEPs based on biological process. (**c**) Clusters of Orthologous Groups of proteins (COG/KOG) category of DEPs including cellular processes and signaling, information storage and processing, metabolism, and poor characterization.

**Figure 4 viruses-14-01383-f004:**
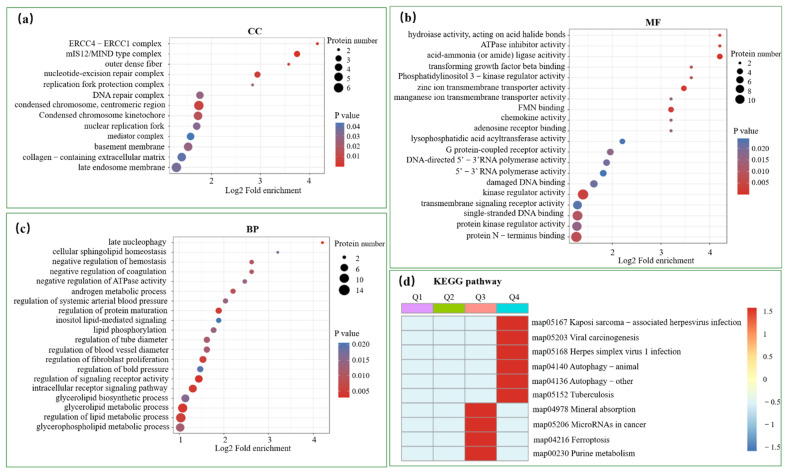
Functional enrichment analysis of DEPs based on GO and KEGG pathway. The vertical axis shows the top 20 enriched biological processes, and the horizontal axis represents the richness factor. The color and size of the dots represent the range of the *p*-value and the number of DEPs mapped to the indicated GO terms (**a**–**c**), respectively. A two-tailed Fisher’s exact test was employed to test the enrichment of the DEPs against all identified proteins. *p*-value < 0.05 was considered significant. (**d**) Analysis of the significant differences in the KEGG pathway showed that DEPs were mostly grouped in the Q3 and Q4 categories. Q Category: Q1 (<0.769), Q2 (0.769–0.833), Q3 (1.2–1.3), and Q4 (>1.3).

**Figure 5 viruses-14-01383-f005:**
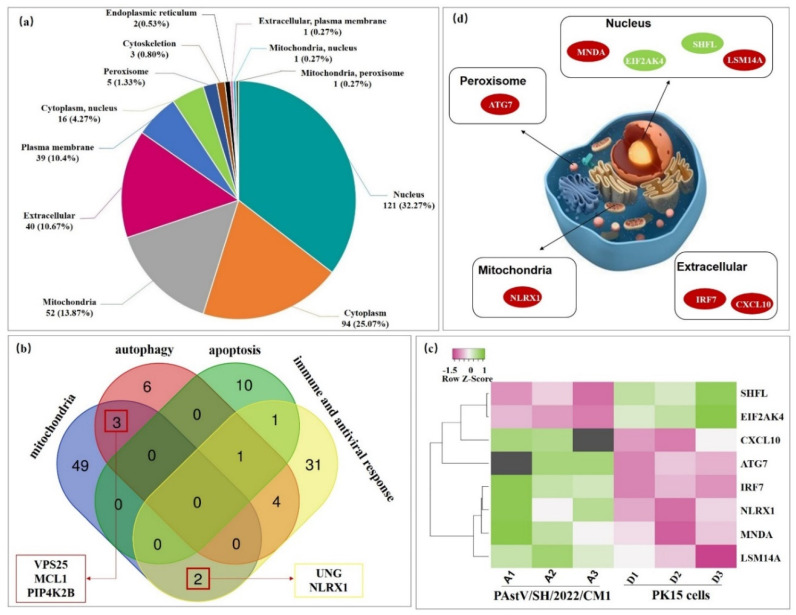
Functional characterization of the DEPs in organelles. (**a**) Subcellular localization analysis of DEPs. (**b**) Venn diagrams show the differentially expressed proteins (DEPs) (shared or unique) between each comparison (http://bioinformatics.psb.ugent.be/webtools/Venn/, accessed on 15 May 2022). (**c**) DEPs that participated in the defense response to the virus were screened and analyzed by Heatmapper (http://www.heatmapper.ca/expression/, accessed on 15 May 2022). Green refers to upregulated DEPs, while pink refers to downregulated DEPs. (**d**) Subcellular location of the DEPs involved in the defense response to the virus.

**Figure 6 viruses-14-01383-f006:**
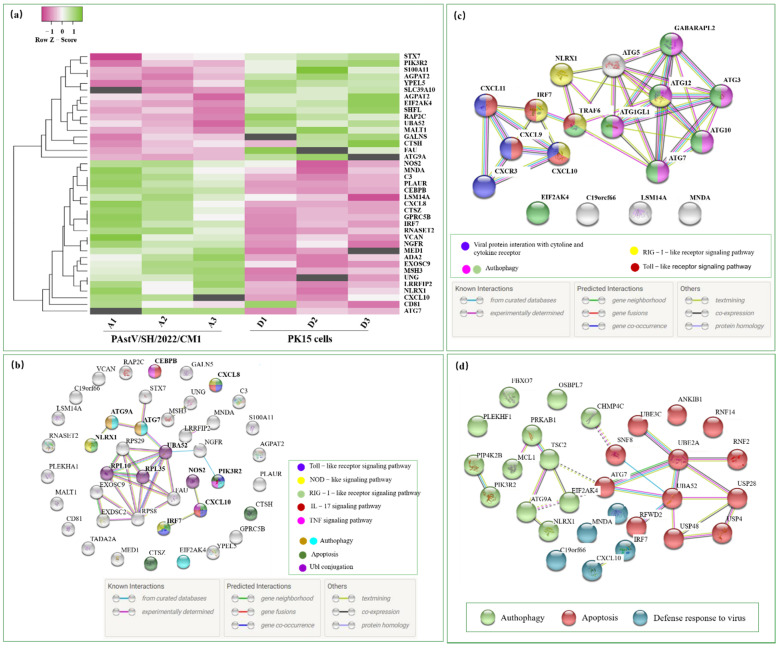
Exploration of the protein–protein interaction (PPI) networks. (**a**) DEPs that participated in the immune and antiviral response were screened and analyzed by Heatmapper. PPI networks of the DEPs involved in the immune and antiviral response (**b**) and defense response to the virus (**c**) were built using the STRING database (https://cn.string-db.org/, accessed on 15 May 2022) with the interaction score set to high confidence (0.700). (**d**) The PPI networks of the DEPs in terms of autophagy, apoptosis, and defense response to the virus were analyzed. DEPs related to autophagy are colored in green and those related to apoptosis in red, while the blue balls represent the DEPs involved in the defense response to virus.

## Data Availability

All data are available in the manuscript and the Appendix A.

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
