# Peer review of "Genomic Divergence Characterization and Quantitative Proteomics Exploration of Type 4 Porcine Astrovirus"

_viruses, 2022, doi:10.3390/v14071383_

Round 1

Reviewer 1 Report

The study "Genomic divergence characterization and quantitative proteomics exploration of type 4 porcine astrovirus" isolated and characterized one strain of porcine astrovirus. The characterization included sequencing, search for genetic recombination, and proteomics of infected PK15 cells.

1- Overall, the manuscript needs some English review: for example, lines 31-32. "causing diseases both asymptomatic and systematic" First, if there is a disease, it cannot be asymptomatic. What is possible is to have an asymptomatic infection. Second, there is no "systematic" disease. Another example: line 80: I do not think magaligned is a word. 

1- The introduction provides little basis for the study and needs to be thoroughly edited. Also, the use of the word novel PAstV is misleading. The authors only identified a virus with limited genetic divergence from known sequences. The use of "novel" is restricted for the discovery of viruses belonging to new species or subtypes...

2- Methods needs more details in all section. More details of the virus amplification, virus origin, source of antibodies and specificity, primers used for sequencing, and how many sequencing replicates.

3- How many replicates of the proteomics characterization were conducted? Why an MOI of 5?

4- Were the full 5UTR and 3UTR recovered? This needs clarification in the manuscript.

5- Most of the figures are impossible to read. Please redo them.

6- Line 239: It sounds like the virus is replicating in the mitochondria. Please rephrase this, and it is likely that every virus will affect the mitochondria metabolism, so I would tone down these findings.

7- The result section presents information that would better fit the discussion. 

8- The discussion is superficial and marginal from the obtained results. The first and fourth paragraphs of the discussion could be mostly omitted. The discussion of the proteomics part needs to be significantly expanded because it falls short of the expectations of this type of work.

9- Supplementary data is not available even after registration to use "Zenodo". The lack of data availability hampers the analyses of part of the methods and findings. Please make data readily available.  

10- Please discuss the possible effect and downsides of using PK15 cells for proteomics, as these cells would not be the virus target in vivo.

Author Response

Point 1: Overall, the manuscript needs some English review: for example, lines 31-32. "causing diseases both asymptomatic and systematic" First, if there is a disease, it cannot be asymptomatic. What is possible is to have an asymptomatic infection. Second, there is no "systematic" disease. Another example: line 80: I do not think magaligned is a word.

Response 1: I’m sorry for the vague description. We have carried out the English editing of this manuscript again.

Point 2: The introduction provides little basis for the study and needs to be thoroughly edited. Also, the use of the word novel PAstV is misleading. The authors only identified a virus with limited genetic divergence from known sequences. The use of "novel" is restricted for the discovery of viruses belonging to new species or subtypes.

Response 2: The preliminary works of our team had been added in the Introduction. And the nonstandard definitions in this manuscript have been revised.

Point 3: Methods needs more details in all section. More details of the virus amplification, virus origin, source of antibodies and specificity, primers used for sequencing, and how many sequencing replicates.

Response 3: Thanks for the suggestions. We have added the important information in Materials and Methods.

Point 4: How many replicates of the proteomics characterization were conducted? Why an MOI of 5?

Response 4: Three replicates of proteomics characterization were conducted. Virus titer (106.3TCID50/ml)/cell numbers (4X105)=MOI (4.988), then a MOI of 5 was used in this study.

Point 5: Were the full 5UTR and 3UTR recovered? This needs clarification in the manuscript.

Response 5: I’m sorry for less preciseness. The complete genome of PAstV/SH/2022/CM1 were sequenced. Relevant information had been supplemented in Materials and Methods.

Point 6: Most of the figures are impossible to read. Please redo them.

Response 6: Thanks for your suggestions. We remade the figures to improve the clarity.

Point 7: Line 239: It sounds like the virus is replicating in the mitochondria. Please rephrase this, and it is likely that every virus will affect the mitochondria metabolism, so I would tone down these findings.

Response 7: Fixed.

Point 8: The result section presents information that would better fit the discussion.

Response 8: Thanks for the suggestion and we have revised it.

Points 9: The discussion is superficial and marginal from the obtained results. The first and fourth paragraphs of the discussion could be mostly omitted. The discussion of the proteomics part needs to be significantly expanded because it falls short of the expectations of this type of work.

Response 9: We have revised it thoroughly.

Point 10: Supplementary data is not available even after registration to use "Zenodo". The lack of data availability hampers the analyses of part of the methods and findings. Please make data readily available.  

Response 10: I’m very sorry and thanks for the friendly warning. We have resubmitted the supplementary data.

Point 11: Please discuss the possible effect and downsides of using PK15 cells for proteomics, as these cells would not be the virus target in vivo.

Response 11: Thanks for the suggestion and we have fixed it in Discussion.

Reviewer 2 Report

In the paper the authors investigated the molecular basis of the infection with  a novel porcine astrovirus (PAstV; PAstV/CH/2022/CM1) pathogenesis and explore the host cell response to PAstV/CH/2022/CM1 infection using proteomics. The results indicate that viral infection elicited protein changes, and the mitochondria seems to be an primary and important target in virus infection. In particular, NOD-like receptor X1 in the mitochondria participated in several important antiviral signaling pathways in PAstV/CH/2022/CM1 infection, which were closely related to mitophagy. The data from this study provide more information for understanding the virus genomic characterization and the potential antiviral targets in PAstV infection. This paper presents the results of a study on porcine astrovirus conducted in vitro with the use of PK15 cell line. This is a fundamental study, which extends the knowledge in the field of pathogenesis of astrovirus infections, conducted in vitro. The results should be confirmed in future in vivo studies. However, astroviruses do not currently appear to be important pathogens of pigs from the clinical point of view.

Minor comments:

Line 29, 33 ect : Astroviruse, viruse  …it should be corrected

Author Response

Point 1: In the paper the authors investigated the molecular basis of the infection with  a novel porcine astrovirus (PAstV; PAstV/CH/2022/CM1) pathogenesis and explore the host cell response to PAstV/CH/2022/CM1 infection using proteomics. The results indicate that viral infection elicited protein changes, and the mitochondria seems to be an primary and important target in virus infection. In particular, NOD-like receptor X1 in the mitochondria participated in several important antiviral signaling pathways in PAstV/CH/2022/CM1 infection, which were closely related to mitophagy. The data from this study provide more information for understanding the virus genomic characterization and the potential antiviral targets in PAstV infection. This paper presents the results of a study on porcine astrovirus conducted in vitro with the use of PK15 cell line. This is a fundamental study, which extends the knowledge in the field of pathogenesis of astrovirus infections, conducted in vitro. The results should be confirmed in future in vivo studies. However, astroviruses do not currently appear to be important pathogens of pigs from the clinical point of view.

Response 1: Thank for the review. Our research team has focused on the epidemiological surveillance of porcine diarrhea pathogens for more than ten years. We found that he infection patterns of diarrhoeal pathogens have changed greatly. Porcine epidemic diarrhea virus (PEDV), transmissible gastroenteritis virus (TGEV) and porcine rotavirus (PoRV) were the three dominanted pathogens in diarrheal disease ten years ago. Nevertheless, the prevalence of several underappreciated diarrheic pathogens, such as PAstV, porcine sapello virus (PSV), porcine kobuvirus (PKoV),are increasing now while that of TGEV and PoRV have dwindled. Importantly, these underappreciated diarrheic pathogens often co-infect with PEDV in clinical samples, and plays synergistic pathogenic roles. This indicates that we need to pay attention to the pathogenic mechasim of these pathogens. Morever, PAstV has the possibility of interspecies transmission , which is of great public health significance. So we consider PAstV to be one of the most promising diarrheic pathogens except PEDV.

Point 2: Minor comments: Line 29, 33 ect : Astroviruse, viruse  …it should be corrected.

Response 2: Thanks very much and we have revised.

Round 2

Reviewer 1 Report

Thanks for the conducted editions.